# Avocado Intake, and Longitudinal Weight and Body Mass Index Changes in an Adult Cohort

**DOI:** 10.3390/nu11030691

**Published:** 2019-03-23

**Authors:** Celine Heskey, Keiji Oda, Joan Sabaté

**Affiliations:** Center for Nutrition, Healthy Lifestyle and Disease Prevention, 24951 North Circle Drive, School of Public Health, Loma Linda University, Loma Linda, CA 92350, USA; koda@llu.edu (K.O.); jsabate@llu.edu (J.S.)

**Keywords:** avocado, obesity, fat, body mass index, adiposity, weight

## Abstract

Avocados contain nutrients and bioactive compounds that may help reduce the risk of becoming overweight/obese. We prospectively examined the effect of habitual avocado intake on changes in weight and body mass index (BMI). In the Adventist Health Study (AHS-2), a longitudinal cohort (~55,407; mean age ~56 years; U.S. and Canada), avocado intake (standard serving size 32 g/day) was assessed by a food frequency questionnaire (FFQ). Self-reported height and weight were collected at baseline. Self-reported follow-up weight was collected with follow-up questionnaires between four and 11 years after baseline. Using the generalized least squares (GLS) approach, we analyzed repeated measures of weight in relation to avocado intake. Marginal logistic regression analyses were used to calculate the odds of becoming overweight/obese, comparing low (>0 to <32 g/day) and high (≥32 g/day) avocado intake to non-consumers (reference). Avocado consumers who were normal weight at baseline, gained significantly less weight than non-consumers. The odds (OR (95% CI)) of becoming overweight/obese between baseline and follow-up was 0.93 (0.85, 1.01), and 0.85 (0.60, 1.19) for low and high avocado consumers, respectively. Habitual consumption of avocados may reduce adult weight gain, but odds of overweight/obesity are attenuated by differences in initial BMI values.

## 1. Introduction

The current prevalence of overweight and obesity in the United States (U.S.) is 70.7% and 37.9%, respectively, among adults [1]. Worldwide, there is a trend towards an increasing prevalence of overweight and obesity [2]. Considering that excess adiposity and adult weight gain contribute significantly to current trends in morbidity and mortality, especially the prevalence of cardiovascular disease, type 2 diabetes, and certain cancers [3,4,5,6], it is imperative that effective methods for reducing adult weight gain are identified.

Nutrient-dense, whole food choices may help to abate adult weight gain and reduce the risk of overweight or obesity. Avocados, a nutrient-dense and medium-caloric-dense whole food [7,8,9], may help to reduce the risk of weight gain and excess adiposity. Benefits of avocados may be attributed to various components including dietary fiber, phytochemicals, mannoheptulose, and monounsaturated fatty acids (MUFAs) [7,8,9,10,11]. Dietary fiber may reduce the risk of weight gain due to various mechanisms including improving satiety, and inhibition of fat absorption [12,13]. Mannoheptulose, a monosaccharide found in avocados, may also have an impact on calorie intake and weight loss, via changes to gut hormones and energy expenditure [9,11,14,15]. Diets rich in MUFAs have been found to prevent the gain of fat-mass, or contribute to loss of excess adiposity [16,17,18]. There is evidence to suggest that avocados may improve satiety, and reduce hunger and food consumption [19], possibly impacting gut hormones [20], which would ultimately assist with calorie intake and weight management.

There are very few studies that have examined the relationship between avocado intake and adiposity. Animal studies indicate a trend towards lower body weight related to avocado administration [12,13,21]. Of the few human intervention studies that have been conducted, the findings are mixed [22,23,24,25]. Researchers of a cross-sectional National Health and Nutrition Examination Survey (NHANES) study reported that subjects who consume avocados had significantly lower BMI, body weight, and waist circumference than non-consumers [7]. No other large epidemiological studies have been used to analyze the longitudinal relationship between avocado intake, and changes in weight or obesity risk. Therefore, the objective of this study is to examine the effect of habitual avocado intake on adult weight gain, and changes in body mass index (BMI).

## 2. Materials and Methods

This was a longitudinal analysis on changes of weight and BMI in subjects from the Adventist Health Study-2 (AHS-2) cohort (registered at Clinicaltrials.gov: NCT03615599). This cohort is comprised of approximately 96,000 members (~65% female; ~27% black), who at the time of enrollment, resided in the U.S., and Canada [26]. Due to several unique characteristics, there are some advantages in examining diet and health outcomes in the AHS-2 cohort. Subjects in the cohort have very low rates of smoking (1.1%) and alcohol consumptions (6.6%) [26]. Dietary intake varies considerably, with approximately 4.2%, 31.6%, 11.4%, and 52.9% of subjects being classified as vegan, lacto-ovo vegetarian, pesco-vegetarian, and non-vegetarian, respectively [26]. Additional details on characteristics of the cohort are published elsewhere [26]. Enrollment for the AHS-2 study occurred from February 2002 to December 2007. Eligible subjects included individuals who were proficient in English, and ≥30 years of age, who completed a comprehensive lifestyle questionnaire [26]. This baseline questionnaire was used to collect data on self-reported height and weight, date of birth, gender, ethnicity, health history, education, physical activity, use of tobacco, and typical dietary intake. Self-reported physical activity was previously validated in Adventists [27,28].

Collection of follow-up data is ongoing and includes information about self-reported weight, and changes in health history. The study was conducted in accordance with the Declaration of Helsinki, and the Institutional Review Board of Loma Linda University approved the AHS-2 cohort study. Subjects gave informed consent to participate in the AHS-2 cohort study.

Dietary intake assessment: The comprehensive lifestyle questionnaire included a 204-item quantitative food frequency questionnaire (FFQ), which was used to assess dietary intake [29]. Avocado intake was assessed with one item in the FFQ: subjects were asked to recall the average frequency and amount of avocado and/or guacamole intake in the last year. Choices for frequency of intake included ‘never or rarely’, ‘1–3 times per month’, ‘one time per week’, ‘2–4 times per week’, ‘5–6 times per week’, ‘one time per day’, and ‘two or more times per day’. Choices for serving sizes included a standard serving (¼ of a medium sized avocado or ¼ cup of avocado or guacamole); ≤½ of the standard serving; and ≥1 ½ of the standard serving. The standard size of avocado in grams in our database is 32.02 g. Daily avocado intake (g/day) was calculated using the product-sum method: f × s × *n* where f = the weighted frequency of avocado; s = the weighted portion size of avocado; and *n* = standard serving size of avocado (grams). Daily avocado intake was categorized into the following groups: 1) consumers versus non-consumers; and 2) non-consumers, low (>0 to <32 g/day), and high (≥32 g/day) consumers.

Validity of avocado intake was assessed in a calibration sub-study (representative sample of ~1000 subjects from AHS-2), comparing intake estimated from a FFQ with intake calculated from 24-h dietary recalls. The mean de-attenuated correlation between the FFQ and 24-h dietary recalls for avocado intake was 0.52 and 0.50 for white and black subjects respectively [30]. Data from the questionnaire was also used to estimate total energy intake, and determine dietary patterns, which have been previously defined [31]. We further categorized subjects into vegetarian or nonvegetarians. Avocado intake varied by dietary pattern within the AHS-2 cohort [31], and prior research indicates a relationship between avocado intake and diet quality [7].

Anthropometric data assessment: The comprehensive lifestyle questionnaire included questions regarding self-reported height and weight. The question to assess height and weight was ‘what is your current height and weight?’ with both write-in and circle-in-weight response options. This data was used to calculate baseline BMI as kg/m^2^. Self-reported height and weight were validated in the calibration sub-study. Weight and height were measured during calibration sub-study clinic visits, the details of which were reported previously [32]. The correlation between self-reported and measured height and weight were 0.96 and 0.96, respectively [32].

Two follow-up questionnaires were sent to collect information on current self-reported weight [30,33], which will be referred to as follow-up weight 1 (Wt1) and weight 2 (Wt2). These questionnaires were initially sent out in 2011 and 2013, meaning that the follow-up time varies from 4 to 11 years. Of the subjects with available Wt1 data, 14% are missing Wt2. This data, along with height data collected at baseline, were used to calculate follow-up BMI values. Weight change was also calculated for both follow-up periods (from baseline). Overweight was defined as a BMI ≥ 25 to <30 kg/m^2^, and obesity as a BMI ≥ 30 kg/m^2^.

Statistical analysis: Subjects were excluded from the analysis if calculated energy consumption was <2092 KJ or >18,828 KJ per day (standard exclusion for implausible energy intake data for AHS-2 analyses); baseline BMI was <18 kg/m^2^ or >40 kg/m^2^; if they reported being pregnant at baseline (not assessed during follow-up in an older cohort); and/or indicated they were currently smoking at baseline. Subjects were also excluded for implausible anthropometric values or changes. These exclusions were based on the distribution of the samples, and clinical judgement. Female subjects were excluded if they reported a height <142 cm or >183 cm, and male subjects were excluded if they reported a height <152 cm or >198 cm. Subjects were also excluded if there were implausible changes in weight. These exclusions reduced the sample size by 13,766 subjects. Subjects were also excluded due to missing questionnaire return dates (*n* = 2), and missing avocado intake (*n* = 5547). These exclusions reduced the baseline sample size to 77,154 subjects. The analytical sample was further reduced to 55,407 subjects due to missing covariates (gender (*n* = 22), race (*n* = 815), age (*n* = 81), education level (*n* = 910), physical activity (*n* = 9162), sedentary time (*n* = 7842), dietary patterns (*n* = 12,367)). Attrition bias analyses did not indicate differences between subjects with or without missing data.

Descriptive analyses were done for variables of interest. The analyses included calculation of means (SD), or medians (IQR). Marginal means of baseline BMI were calculated after adjusting for age, gender, race, and energy intake. Two-sample t-tests were used to assess the differences between avocado consumers and non-consumers for normally distributed continuous variables, whereas the Mann-Whitney test was used as a nonparametric test for other continuous variables. One-way analysis of variance (ANOVA) was used to assess the differences between avocado non-consumers, low consumers, and high consumers for normally distributed continuous variables. The Kruskal–Wallis test was used as a nonparametric test for other variables. Chi-square analysis was used to assess differences between avocado consumers and non-consumers, and between non-consumers, low consumers, and high consumers for categorical variables. Chi-square analysis was also used to assess differences in the percent change of BMI between baseline and follow-up.

To assess the relationship between changes in BMI and weight over time, and avocado intake, repeated measures data were analyzed using the generalized least squares (GLS) approach. Time (in years), between baseline and follow-up, was calculated for each subject, and used as a continuous variable in the GLS model. For the covariance pattern among repeated measures, the exponential covariance structure was found to be most appropriate for the data. This covariance structure was used in all longitudinal models for weight and BMI. The two dependent variables, weight and BMI, were log-transformed prior to analysis to achieve approximate normality. The main exposure of the model, daily avocado intake (g/day), was also log-transformed (log_e_(X + 1) where X is avocado intake) to dampen the influence of outliers. Daily avocado intake was also adjusted for total energy intake, using the residual method [34]. The model also included an interaction term between avocado intake and time, to examine if the rate of change differs according to avocado intake. Additional covariates were included in the model to account for confounding by (1) gender; (2) race (black, non-black); (3) age (age at baseline), (4) energy intake; (5) socioeconomic status: education level (high school or less, some college, college graduate or more), (6) activity level: physical activity (hours/day), and sedentary time (hours/day); and (7) diet quality: dietary pattern (vegetarian, nonvegetarian). Covariates were identified based on statistical and/or biological connections to the exposure and outcomes.

For ease of interpretation, the GLS models were used to estimate weight and BMI at baseline, and five years after baseline for avocado intake at 0 g/day and 32 g/day. Percent changes from baseline were calculated based on these estimated values.

In order to assess the impact of certain factors on changes in weight and BMI over time, subgroup analysis was also done for age (categorized as <60 years and ≥60 years); baseline BMI (normal, overweight, and obese); gender (male and female); race (black and non-black); and dietary patterns (vegetarian and nonvegetarian). GLS models were also used to estimate weight and BMI at baseline, and 5 years after baseline for avocado intake at 0 g/day and 32 g/day for the subgroup analyses.

Models were used to calculate odds ratios (OR) (95% CI) of becoming overweight or obese (outcome events of BMI ≥25 kg/m^2^ calculated from Wt1 and Wt2) based on the level of avocado intake (none, low, and high), in those who had a normal BMI at baseline. The marginal models were fit using the alternating logistic regression (ALR) algorithm [35] to account for the association between repeated measurements (Wt1 and Wt2) with log odds ratios. Since we restricted this analysis to those with normal BMI at baseline, in the marginal model only measurements from Wt1 and Wt2 were used to avoid complete separation at baseline. Covariates included follow-up time, gender, race, age at baseline, total energy intake, education level, physical activity, sedentary time, dietary pattern, and baseline BMI. Estimated beta coefficients from the marginal logistic model and their robust standard error estimates were used to calculate adjusted ORs associated with avocado intake and their 95% CIs.

Statistical analyses were done using SAS 9.4 software (SAS Institute Inc., Cary, NC, USA).

## 3. Results

### 3.1. Descriptive Results

The average age (SD) of the analytical sample at baseline was 55.9 (13.7) years. The average baseline weight (SD) and BMI (SD) were 76.0 (15.9) kg and 26.6 (4.7) kg/m^2^, respectively. The proportion of vegetarians (including vegans) was 37.8%. Additional baseline descriptive data can be found in Table 1. The average (SD) follow-up times between baseline data collection, and return dates for the follow-up weights were Wt1: 7.4 (1.2) years, and Wt2: 9.1 (1.3) years. Average weight and BMI at Wt1 were 75.4 kg and 26.4 kg/m^2^, respectively, and for Wt2 were 75.1 kg, and 26.3 kg/m^2^ respectively.

In terms of frequency of intake, approximately 41% of individuals reported never or rarely consuming avocado, and 34% of subjects consumed avocado occasionally (at least 1–3 times per month). Regular consumers (consuming avocado at least once per week) made up ~25% of subjects. Of those consuming avocado, 69% consumed the standard serving size of avocado, while 18% consumed less and 13% consumed more.

Median avocado intake among consumers was 2.3 g/day, with a range of 1.1 to 120.1 g/day. Avocado consumers, on average, had a significantly higher energy (SD) intake than non-consumers (7535.8 (3063.9) versus 8261.7 (3025.0) KJ/day). Consumers, however, had a lower average weight (SD) (77.9 (16.2) versus 74.5 (15.5) kg; *p* < 0.0001) and BMI (SD) (27.3 (4.8) versus 26.0 (4.5) kg/m^2^; *p* < 0.0001), and were less likely to be sedentary, but more likely to be vigorously active than non-consumers.

When comparing zero, low (>0 to <32 g/day), and high avocado intake (≥32 g/day), age, caloric intake, and sedentary hours per day differed significantly (Table 1). Weight and BMI also differed by zero, low, and high avocado intake (Table 2). Baseline weight and BMI followed a declining trend as avocado intake increased (Table 2).

In terms of follow-up weights, average BMI at Wt1 was 27.1 kg/m^2^, 26.0 kg/m^2^, and 24.7 kg/m^2^ for avocado non-consumers, low, and high consumers respectively. Similarly, at Wt2, average BMI was 27.0 kg/m^2^, 25.9 kg/m^2^, and 24.5 kg/m^2^ among non-consumers, low and high consumers, respectively.

Among subjects who had a normal BMI at baseline, approximately 16.7% and 16.6% of became overweight/obese by Wt1 and Wt2, respectively. In assessing the impact of avocado intake on the conversion to overweight/obesity, we found that avocado non-consumers were more likely to become overweight/obese compared to consumers. For Wt1, 18.7%, 15.8%, and 10.5% of non-consumers, low, and high consumers became overweight/obese respectively (*p* < 0.0001). For Wt2, 19.0% of non-consumers, 15.6% of low consumers, and 10.2% of high consumers converted to being overweight/obese (*p* = 0.004).

Alternatively, among those subjects who were overweight or obese at baseline, 11.7%, 12.7%, and 12.8% of non-consumers, low, and high consumers respectively, became normal weight Wt1 (*p* = 0.04), and 13.4%, 14.7%, and 17.6% of non-consumers, low, and high consumers respectively, became normal weight by Wt2 (*p* = 0.004).

### 3.2. Results of Mixed Models Analyses

Among individuals with a normal BMI at baseline, avocado consumers’ weight and BMI increased at a lower rate than non-consumers. This translates to an increase in kg of 0.79% over five years for non-consumers versus 0.26% for high avocado consumers (≥32 g/day). For a 75 kg individual this translates to a difference of 0.4 kg in weight change over 5 years between non-consumers and high avocado consumers. We did not find significant changes in weight and BMI among those who were overweight or obese at baseline. Among older subjects in the AHS-2 cohort, there is a tendency to lose weight over time. We found that avocado consumers ≥60 years of age, had less of a tendency to lose weight and BMI over time compared to non-consumers. We did not find significant results for other subgroup analyses (gender, race, dietary pattern). Sensitivity analyses including residential region as a covariate did not substantially change the results. See Table 3 for results of the mixed model analysis on weight.

### 3.3. Results of Logistic Regression Analyses

Among subjects of normal weight at baseline, those who consumed avocado had a lower odds of becoming overweight or obese during follow-up (Table 4). The odds (95% CI) were lowered in a stepwise fashion with increasing intake of avocado: 0.89 (0.82, 0.96) for low consumers, and 0.61 (0.44, 0.85) for high consumers, compared to non-consumers (reference) after adjusting for follow-up time, energy intake, gender, race, age, education, physical activity, sedentary time, and dietary patterns. Further analysis adjusting additionally for baseline BMI attenuated the findings: 0.93 (0.85, 1.01) for low consumers, and 0.85 (0.60, 1.19) for high consumers, compared to non-consumers (reference). Adding residential region as a covariate was insignificant and did not substantially change the results.

## 4. Discussion

We found that, among avocado consumers in the AHS-2 cohort, there was a reduction in the odds of becoming overweight/obese compared to those who did not eat avocado, but this finding was attenuated by adjusting for baseline BMI. Differences in baseline BMI had more of an impact on the odds of becoming overweight/obese than differences in avocado intake. These results may also be partly explained by the relatively small difference in weight or BMI changes between avocado consumers and non-consumers.

Utilizing data from one 24-h dietary recall, collected during several cycles of NHANES (2001–2008), Fulgoni and colleagues reported average (SD) avocado intake among avocado consumers (*n* = 347) of 70.1 (5.4) g/day [7]. This is quite a contrast to our findings, that average avocado consumption among consumers in the AHS-2 cohort is 2.3 g/day. One reason for this difference may be due to the difference in methods used to measure avocado intake. The AHS-2 FFQ was designed to estimate intake over one year, or habitual intake. Data from one 24-h dietary recall that was extrapolated from NHANES, is most likely not reflective of habitual avocado intake. Additionally, whereas ~2% of the NHANES subjects were avocado consumers [7], ~58% of AHS-2 subjects habitually consume avocado. We are also limited by the fact that the highest amount of avocado intake that can be calculated from our FFQ is as a result of the maximum weighted frequency (2.5 per week), and serving size (1.5 times the standard) available for subjects to choose from. By utilizing another method of dietary assessment, it is possible that average habitual intake is possibly even higher.

Despite these differences, the findings in our study correspond with what Fulgoni and colleagues have reported. They reported that NHANES avocado consumers had significantly lower weight (78.1 kg versus 81.1 kg) and BMI than non-consumers (26.7 kg/m^2^ versus 28.4 kg/m^2^) [7]. Similarly, we found that AHS-2 avocado consumers had significantly lower weight (74.5 kg versus 77.9 kg), and BMI (26.0 kg/m^2^ versus 27.3 kg/m^2^) than non-consumers. Our findings confirm that habitual avocado intake, of fairly minimal average amounts, are associated with lower weight and excess adiposity cross-sectionally. Our longitudinal analyses indicate that among subjects who had a normal BMI at baseline, avocado consumers gained less weight and BMI over time than non-consumers. The differences however are small, and for individuals who were normal weight at baseline, the results are attenuated by differences in baseline BMI.

We found a declining trend in average weight during the follow-up for the entire AHS-2 cohort. This trend has been noted previously in the AHS-2 cohort [33], and in other cohort studies [4]. Adams et al. reported weight loss in subjects 50 to 69 years of age in the National Institutes of Health American Association of Retired Persons (NIH-AARP) study [4]. Taking these factors into consideration, we did stratified analyses for age, and among those ≥60 years of age, we found a significant association between avocado intake, and changes in weight and BMI: subjects who consumed avocado had a lower reduction in their weight than those who did not consume avocados. This may actually be a beneficial outcome considering that weight loss in elderly individuals is associated with poor health outcomes, and deleterious changes in body composition [4]. There is evidence that minimal weight loss (>0.2 kg/year) in older individuals (>50 years of age) may increase risk of mortality [4].

Various mechanisms, related to the nutrients and bioactive compounds in avocados, may help to explain some of the findings related to changes in weight, including impacts on satiety, metabolism, and gut microbiota. Avocado extract has been found to impact the expression of genes involved in fat metabolism and appetite in animals (i.e., fatty acid synthase, fibroblast growth factor 21, leptin, lipoprotein lipase) [21]. A few animal studies provide evidence that mannoheptulose may favorably impact gut hormones (i.e., ghrelin, glucagon like peptide-1), and energy expenditure [14,15]. Our analysis of the impact of avocado intake on gut hormones in humans was not as conclusive, however [20]. The beneficial effect of MUFA on weight may be mediated through its effect on the stearoyl-CoA desaturase 1 enzyme [17].

Avocados are considered to be a rich source of dietary fiber, which is known to increase satiety [19]. We have previously reported that subjects fed an avocado test meal reported significantly higher post-meal satisfaction and lower desire to eat, compared to the control test meal period [19]. Dietary fiber ferments in the gut to form short-chain fatty acids (SCFAs) which serve directly as a substrate for gut microbiota [8,36]. Additionally, SCFAs, particularly butyrate, have been found to have an impact on gut hormones (i.e., leptin, glucagon like peptide-1), which affect appetite, and weight [36]. SCFAs also impact genes that are responsible for deposition of fat [36]. Unabsorbed sugars, also serve as substrates to intestinal bacteria [36], so it may be hypothesized that unabsorbed mannoheptulose may have an impact. A clear benefit of MUFAs on gut microbiota profile is promising but not fully elucidated [37]. Only one study has been done to evaluate the effect of avocado intake on gut microbiota: a 6-week study in rats revealed that avocado intake is associated with overall increases in SCFAs, particularly, acetate [8].

Fulgoni et al. reported that avocado consumers had a significantly higher diet quality than non-consumers [7]. It may be hypothesized that avocado, as an indicator of diet quality, may be associated with reduced risk of age-related declines in lean body mass. It would be useful to examine the effect of avocado consumption on changes in body composition of older individuals.

Strengths of our study include the size of the sample, and the fact that the analyses were longitudinal. Additionally, more than half of our cohort reported habitual avocado consumption. Avocado intake was quite varied, which makes this a useful cohort to examine the health effects related to intake. Due to the unique health habits espoused by most of the cohort members, very few smoke, or drink alcohol, which limits confounding by these factors when examining the relationship between diet and health. These unique health habits may however somewhat limit the generalizability of the results.

Limitations of this study include use of self-reported anthropometric measurements and dietary intake method. As validation of these measurements suggest that these methods are good to excellent approximations of reference standard measurements [30,32], we feel that these limitations did not present significant issues in the analyses. Another issue is that we did not have a follow-up measurement of dietary intake, so we were unable to assess changes in avocado intake over time. It is also possible that there was residual confounding by dietary factors other than kcals and dietary patterns. It should be noted that there was very minimal change in weight over the course of follow-up in our cohort. Considering that part of our population is elderly, there was an overall weight change trend of loss instead of gain. It would be useful to repeat this study in a cohort of younger individuals at risk for overweight and/or obesity. Despite these limitations, we were still able to find significant associations between avocado intake, and weight and BMI. 

## 5. Conclusions

In conclusion, habitual avocado intake is associated with a lower prevalence of excess weight, and attenuates adult weight gain in normal weight individuals over time in this health oriented population. Higher amounts of habitual avocado intake are associated with lower odds of becoming overweight and/or obese, but this is attenuated by differences in baseline BMI. While the clinically the changes in weight are minor, there are possible overall public health implications of long-term changes in weight at the population level. 

## Figures and Tables

**Table 1 nutrients-11-00691-t001:** Baseline characteristics of Adventist Health Study-2 subjects along with differences by level of avocado intake ^1^.

Characteristics	Categories	All	Non-Consumers of Avocado	Low Avocado Intake ^2^	High Avocado Intake ^2^
Number of subjects		55,407	23,242	31,346	819
Age ^3^, years		55.9 (13.7)	55.6 (13.9) *	56.1 (13.5)	58.1 (13.8)
Caloric intake ^3^, KJ/day		7957.1 (3062.7)	7535.8 (3063.9) *	8218.2 (3001.2)	9930.3 (3439.7)
Avocado Intake ^4^, g/day		2.2 (3.2)	0	2.3 (2.4)	37.8 (16.0)
Avocado Intake ^4^, energy adjusted, g/day		1.6 (3.4)	0	2.8 (3.3)	34.9 (12.0)
Sedentary ^4,5^, hours/day		2.0 (2.5)	2.4 (2.6) *	2.0 (2.6)	1.6 (2.3)
Vigorous activity ^4,5^, hours/day		0.2 (0.8)	0.2 (0.8) *	0.3 (0.8)	0.4 (1.3)
Gender **^6^ (%)	Female	62.7	62.0	63.2	65.7
	Male	37.3	38.0	36.8	34.3
Race *^6^ (%)	Black	21.7	30.8	15.1	16.7
	Non-Black	78.3	69.2	84.9	83.3
Education *^6^ (%)	≤High school	17.8	23.5	13.6	16.4
	Some college	38.9	40.7	37.6	40
	≥College graduate	43.2	35.8	48.8	43.6
Dietary Pattern *^6^ (%)	Nonvegetarians	62.2	70.6	56.5	42.4
	Vegetarians	37.8	29.4	43.5	57.6

^1^ Mean (SD) unless otherwise stated. ^2^ Low avocado intake: >0 to <32 g/day; High avocado intake: ≥32 g/day. ^3^ One-way ANOVA used to test differences between non-consumers, low, and high avocado intake. ^4^ Median (IQR). ^5^ Kruskal–Wallis test used to test differences between non-consumers, low, and high avocado intake. ^6^ Chi-square test used to test differences between non-consumers, low, and high avocado intake. * Significantly different from consumers (*p* < 0.0001). ** Significantly different from consumers (*p* < 0.01).

**Table 2 nutrients-11-00691-t002:** Baseline weight and BMI by level of avocado intake in the Adventist Health Study-2 cohort ^1^.

Anthropometric Data	Non-Consumers of Avocado	Low Avocado Intake ^2^	High Avocado Intake ^2^
Weight ^3^, kg	77.9 (16.2) *	74.6 (15.5)	70.5 (14.6)
BMI ^3^, kg/m^2^	27.3 (4.8) *	26.1 (4.5)	24.8 (4.4)
BMI ^4^, kg/m^2^	27.2 (27.1, 27.2) *	26.1 (26.1, 26.8)	24.7 (24.4, 25.0)

^1^ Mean (SD) unless otherwise stated. ^2^ Low avocado intake: >0 to <32 g/day; high avocado intake: ≥32 g/day. ^3^ One-way ANOVA used to test differences between non-consumers, low, and high avocado intake. ^4^ Estimated marginal means (95% CI) adjusted for age, gender, race, and caloric intake. * Significantly different from consumers (*p* < 0.0001).

**Table 3 nutrients-11-00691-t003:** Effect of avocado intake on change in weight and BMI over time in Adventist Health Study-2 cohort ^1^.

Criteria	Categories	Weight β-Coefficient (SE)	% Weight Change over 5 Years		BMI β-Coefficient (SE)	% BMI Change over 5 Years	
			Non-Consumers	High Avocado Intake ^2^		Non-Consumers	High Avocado Intake
All subjects ^3^		0.001 (0.005)	−0.06	−0.04	0.001 (0.005)	0.05	0.07
Age ^3^	<60 years of age	−0.007 (0.006)	1.11	0.98	−0.007 (0.006)	1.22	1.09
	≥60 years of age	0.022 (0.008) ***	−1.97	−1.59	0.022 (0.008) ***	−1.86	−1.47
Baseline BMI ^3^	Normal	−0.304 (0.069) *	0.79	0.26	−0.031 (0.007) *	0.90	0.36
	Overweight	−0.029 (0.089)	−0.27	−0.32	−0.001 (0.008)	−0.19	−0.21
	Obese	−0.043 (0.138)	−1.03	−1.11	−0.005 (0.013)	−0.95	−1.03
Gender ^4^	Female	0.004 (0.074)	0.03	0.04	0.001 (0.007)	0.14	0.16
	Male	0.017 (0.075)	−0.21	−0.21	0.0001 (0.007)	−0.10	−0.10
Race ^5^	Black	−0.038 (0.147)	0.44	0.37	−0.004 (0.015)	0.56	0.50
	Non-black	0.091 (0.057)	−0.20	−0.03	0.010 (0.006)	−0.09	0.09
Dietary Pattern ^6^	Vegetarian	0.005 (0.008)	−0.71	−0.09	0.002 (0.008)	−0.06	0.02
	Nonvegetarian	0.003 (0.007)	−0.00	0.04	0.002 (0.007)	0.11	0.16

^1^ Percent change in weight and BMI from baseline estimated with generalized least squares models. Weight and BMI were log-transformed prior to analysis. Avocado intake was log transformed prior to analysis [log_e_(X + 1)] where X = avocado intake. β-coefficient for mixed models analyses (interaction between time and avocado intake) is the change in the slop of log transformed weight/BMI over time caused by a 1 unit increment of log transformed avocado intake. ^2^ High avocado intake: ≥32 g/day. ^3^ Adjusted for energy intake, gender, race, age, education, physical activity, sedentary time, and dietary pattern. ^4^ Adjusted for energy intake, race, age, education, physical activity, sedentary time, and dietary pattern. ^5^ Adjusted for energy intake, gender, age, education, physical activity, sedentary time, and dietary patterns. ^6^ Adjusted for energy intake, gender, race, age, education, physical activity, and sedentary time. * *p* < 0.0001: Significant difference between avocado consumers and non-consumers in change of weight or BMI over time. *** *p* < 0.01: Significant difference between avocado consumers and non-consumers in change of weight or BMI over time.

**Table 4 nutrients-11-00691-t004:** Odds (95% CI) of becoming overweight/obese by level of avocado intake in Adventist Health Study-2 cohort ^1^.

	Non-Consumers of Avocado	Low Avocado Intake ^2^	High Avocado Intake ^2^
Baseline BMI ^3^	22.5 (1.7) kg/m^2^	22.3 (1.7) kg/m^2^	21.7 (1.8) kg/m^2^
Unadjusted	1 (reference)	0.82 (0.76, 0.88)	0.54 (0.41, 0.70)
Adjusted ^4^	1 (reference)	0.89 (0.82, 0.96)	0.61 (0.44, 0.85)
Adjusted for Baseline BMI ^5^	1 (reference)	0.93 (0.85, 1.01)	0.85 (0.60, 1.19)

^1^ OR from marginal logistic regression. Weight and BMI were log-transformed prior to analysis. Avocado intake was log transformed prior to analysis [log_e_(X + 1)] where X = avocado intake. ^2^ Low avocado intake: >0 to <32 g/day; high avocado intake: ≥32 g/day. ^3^ Subjects with normal weight at baseline. Mean (SD). ^4^ These analyses were adjusted for follow-up time, energy intake, gender, race, age, education, physical activity, sedentary time, and dietary patterns. ^5^ These analyses were adjusted for follow-up time, energy intake, gender, race, age, education, physical activity, sedentary time, dietary patterns, and baseline BMI.

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
