# Peer review of "Avocado Intake, and Longitudinal Weight and Body Mass Index Changes in an Adult Cohort"

_nutrients, 2019, doi:10.3390/nu11030691_

Round 1
Reviewer 1 Report
This is an interesting and well-written paper on a relevant topic. My main issue is that the results should be explained in some more detail.
Please see below my suggestions for improvement:
- The abstract should provide more information such as sample size, location and age range of participants, and a justification for the 32g/day cut-off.
- l. 63: Some information should be added regarding assessment and validity of self-reported physical activity.
- l. 92: If weight and height were also available as measured in clinics, why were these values not used for analysis? Were they available only for a subset?
- l. 95-96: How many participants had missing Wt2 data?
- l. 102: What was the rationale to use exactly these cut-offs to define plausible energy intake?
- l. 103: Was pregnancy at a follow-up visit no exclusion criterion?
- l. 106/107: Please convert inches to cm.
- l. 101-112: Did the authors investigate potential attrition bias due to these exclusions?
- l. 129-130: What was the rationale to log-transform both outcome and exposure variables? Please take into account how this affects the interpretation of the beta coefficients and state this explicitly, also in the abstract.
- l. 132: Please give a reference for the residual method.
- l. 149-159: Were there any participants who were overweight at Wt1, but normal-weight at Wt2? Were this considered to become overweight in this analysis?
- Table 1: Please show all confounders which were used in your analysis, e.g. also smoking and SES.
- Table 1: Why is there such a high proportion of female participants? Is it equally high in the whole AHS-2 study?
- l. 177-181: These numbers seem not to fit to those from table 1. Please explain the difference.
- l. 192-194: How was this adjustment done? Suggest to add an extra table for these results.
- l. 199-205: Was the average BMI at baseline similar between the three normal-weight groups?
- Table 2 is difficult to understand and at least needs a thorough interpretation in the main text. Additionally, it should be mentioned in the table description if log-transformed outcome and exposure variables were used.
- Also table 3 needs a more details interpretation. Did the follow-up time differ in participants who became overweight/obese vs. not? Was follow-up time taken into account in these analyses? Figure 3 legend seems to indicate yes, while the methods section does not mention this.
- Discussion: May avocado intake be associated with other healthy habits and/or regional variation which are not properly contained in the confounder variables used in the analyses?
- In the spirit of Open Science, I want to encourage the authors to make the data and/or the analysis code publicly available, or to provide a statement in the paper why this is not possible.
Author Response
Response to Reviewer 1 Comments
Point 1: The abstract should provide more information such as sample size, location and age range of participants, and a justification for the 32g/day cut-off.
Response 1: With consideration of the 200 word limit, the following amendments were made to the abstract. It has been amended to include the sample size and location of the cohort, and average age of the sample. Regarding the 32 g/day cut-off for avocado intake, it represents the standard serving amount of avocado/guacamole intake that was defined for the FFQ assessment. Subjects in the ‘high’ (≥32 g/day) therefore represent individuals consuming one or more servings per day of avocado. The standard serving size has been added to the abstract.
Point 2: l. 63: Some information should be added regarding assessment and validity of self-reported physical activity.
Response 2: Physical activity was previously validated in Seventh-day Adventists. References: Singh, P.N.; Tonstad, S.; Abbey, D.E.; Fraser, G.E. Validity of selected physical activity questions in white Seventh-day Adventists and non-Adventists. Med Sci Sports Exerc 1996, 28, 1026-1037.; Singh, P.N.; Fraser, G.E.; Knutsen, S.F.; Lindsted, K.D.; Bennett, H.W. Validity of a physical activity questionnaire among African-American Seventh-day Adventists. Med Sci Sports Exerc 2001, 33, 468-475.) We have amended lines 70-71.
Point 3: l. 92: If weight and height were also available as measured in clinics, why were these values not used for analysis? Were they available only for a subset?
Response 3: Yes, the measured height and weight were only done for participants (n~1000) of the calibration sub-study, which is a representative sample of the parent cohort. The initial mention of the calibration sub-study is clarified (lines 89-90).
Point 4: - l. 95-96: How many participants had missing Wt2 data?
Response 4: Among those who have Wt1 data, about 14% have missing Wt2 data.
Point 5: l. 102: What was the rationale to use exactly these cut-offs to define plausible energy intake?
Response 5: This has been the standard practice in prior analyses using Adventist Health Study (AHS-2) dietary data, so we used to maintain consistency of eliminating subjects who may have errors in data responses.
Point 6: l. 103: Was pregnancy at a follow-up visit no exclusion criterion?
Response 6: We recognized that theoretically including a pregnant subjects at the time of follow-up would lead to misclassification. Nevertheless a large portion of the cohort is beyond reproductive age. Question(s) about pregnancy were not included in the follow-up questionnaires.
Point 7: l. 106/107: Please convert inches to cm.
Response 7: Amended – see lines 115-116
Point 8: l. 101-112: Did the authors investigate potential attrition bias due to these exclusions?
Response 8: We have performed additional analyses to assess potential attrition bias. We did not find relevant differences in avocado intake and BMI between subjects with missing variables of interest versus those without the missing variables. Overall, sociodemographic variables were similar between subjects excluded due to missing variables versus those who were not.
Characteristics | Included (n=55,407) | Excluded (n=21,747) | |
Gender (male), % | 37 | 34 | |
Race (non-Black), % | 22 | 30 | |
Education, % | ≤High-school | 18 | 27 |
Some college | 39 | 40 | |
College grad | 43 | 34 | |
Age (year), mean | 56 | 62 | |
BMI (kg/m2), mean | 26.5 | 26.4 | |
Avocado intake (g/d), median | 1.6 | 1.8 |
Point 9: l. 129-130: What was the rationale to log-transform both outcome and exposure variables? Please take into account how this affects the interpretation of the beta coefficients and state this explicitly, also in the abstract.
Response 9: Distributions of weight, BMI and avocado intake were right-skewed. The natural log transformation helped to achieve approximate normality. Line 141, and amended (lines 139-140).
When both the outcome and exposure variables are log-transformed, the estimated beta coefficient associated with the exposure can be interpreted as a power of change in ratio of x that yields the expected change in ratio of y. Approximately, the beta coefficient is the expected percent increase in y when x increases by one percent. The footnotes for Table 3 and 4 (previously table 2 and 3) were updated.
Point 10: l. 132: Please give a reference for the residual method.
Response 10: A reference has been added, Willett, W. Nutritional epidemiology, Third edition. ed.; Oxford University Press: Oxford ; New York, 2013; pp. ix, 529 pages, on line 142.
Point 11: l. 149-159: Were there any participants who were overweight at Wt1, but normal-weight at Wt2? Were this considered to become overweight in this analysis?
Response 11: The marginal logistic regression used here is a type of repeated measures analysis. If a subject is overweight/obese at Wt1, then this was considered as a case at the time of Wt1. If the same subject is reverted into normal weight at Wt2, then this was considered as a non-case at the time of Wt2. This allows us to estimate ORs of becoming overweight/obese at both time points and see how the odds changes over time.
Point 12: Table 1: Please show all confounders which were used in your analysis, e.g. also smoking and SES.
Response 12: Subjects who are smokers were excluded from the analysis (line 112). Education level was used as a measure of socioeconomic status (line 146) in the analysis and was included in table 1.
Point 13: Table 1: Why is there such a high proportion of female participants? Is it equally high in the whole AHS-2 study?
Response 13: Yes, ~2/3rds of the whole AHS-2 cohort are female subjects, and ~1/3rd are male subjects. Our analytical sample reflects the parent cohort. This cohort was derived from the Seventh-day Adventist church, and disparity in gender is common across all U.S. Christian denominations.
Point 14: l. 177-181: These numbers seem not to fit to those from table 1. Please explain the difference.
Response 14: The avocado intake (g/day) in Table 1 is calculated using the following formula: f * s * n where f = the weighted frequency of avocado; s = the weighted portion size of avocado; and n = standard serving size of avocado (grams). The results in the manuscript (lines 188-190) are specific solely to frequency of avocado intake based on questions on the FFQ (lines 79-81). The results on lines 191-192 are specific solely to the categories of intake amount that are on the FFQ (lines 81-83).
Point 15: l. 192-194: How was this adjustment done? Suggest to add an extra table for these results.
Response 15: Marginal means of BMI were estimated for three different avocado intake groups, by including age and caloric intake (both as continuous) and gender and race (both as binary) as covariates in a linear model. A new table (table 2) has been added to the manuscript to combine baseline anthropometric data from table 1 with the results that were adjusted for in the text.
Point 16: l. 199-205: Was the average BMI at baseline similar between the three normal-weight groups?
Response 16: Among normal weight subjects at baseline, the mean (SD) BMI was similar: 22.5 (1.7) kg/m2, 22.3 (1.7) kg/m2, and 21.7 (1.8) kg/m2 for nonconsumers, low, and high avocado consumers respectively. This has been added to table 4 (previously table 3).
Point 17: Table 2 is difficult to understand and at least needs a thorough interpretation in the main text. Additionally, it should be mentioned in the table description if log-transformed outcome and exposure variables were used.
Response 17: The text was amended with a more thorough interpretation of the results in table 3 (previously table 2) – see lines 230-239.
The footnote for table 3 was amended to include ‘Weight and BMI were log-transformed prior to analysis. Avocado intake was log transformed prior to analysis [loge(X+1)] where X = avocado intake.’
Point 18: Also table 3 needs a more details interpretation. Did the follow-up time differ in participants who became overweight/obese vs. not? Was follow-up time taken into account in these analyses? Figure 3 legend seems to indicate yes, while the methods section does not mention this.
Response 18: Yes, the analyses include an adjustment for follow-up time (lines 135, 143, 166). The text referring to Table 4 (previously table 2) results has been amended (lines 260-264). The footnotes for tables 3 and 4 (previously tables 2 and 3) have been updated.
Point 19: Discussion: May avocado intake be associated with other healthy habits and/or regional variation which are not properly contained in the confounder variables used in the analyses?
Response 19: We adjusted for health habit variables (physical activity, sedentary time, differences in dietary patterns) available in our database that have been shown previously to be related to other outcomes that have been studied in this cohort. We acknowledge that subjects in California eat more avocado on average than outside of California (2.58 g/day versus 1.12 g/day). We did not find a substantial difference in BMI between those subjects in California than those residing outside the state (26.3 kg/m2 versus 26.7 kg/m2). We did sensitivity analyses on the mixed models analyses and logistic regression analyses with the addition of a region covariate variable which did not change the results. This has been clarified in the results, lines 238-239, and 263-264.
Point 20: In the spirit of Open Science, I want to encourage the authors to make the data and/or the analysis code publicly available, or to provide a statement in the paper why this is not possible.
Response 20: The data set that we used for our research is from the AHS-2 cohort and the data is not currently available on an external repository. However, AHS has guidelines here: https://publichealth.llu.edu/adventist-health-studies/researchers for how to access the data for collaborative research.
The authors were not sure where to include clarification regarding data access within the manuscript. To the editors, please note the addition to lines 370-371, and let us know if this is an appropriate location.
Reviewer 2 Report
This interesting manuscript examines the association of avocado consumption and weight gain over a period of 4-11 years. Overall, the manuscript is well-written and the messaging is clear. The following comments are will hopefully help to improve the final product.
Title: The title is misleading. Participants with obesity are excluded from analyses and the overall aim of the study is more concerned with longitudinal fluctations in weight.
Introduction: Nice overview of implications of avocado consumption for weight management.
Method: The Adventist Health Study – 2 is not well-explained. Seventh Day Adventists hold unique dietary beliefs that likely differentiates this group from the general population. For example, most Adventists do not eat pork and many are vegetarians. These differences are not acknowledged in the Method and only vaguely mentioned in the discussion.
Participants with obesity were excluded. Why?
Results: Appropriate analytic plan. Tables are somewhat cumbersome – perhaps authors could improve readability by not centering cells.
Discussion: Are there regional differences to consider? For example, if a preponderance of participants are from California (where avocados may be more available and rates of overweight/obesity are lower), this is an important confounder to consider.
Other issues noted:
Line 66 – subject should be plural
Line 68 – Is ‘dietary intake assessment’ a section heading?
Line 88 – section heading?
Line 101 – section heading?
Line 123 – section heading?
Line 149 – section heading?
Line 334 – y on ‘Health’
Author Response
Response to Reviewer 2 Comments
Point 1: Title: The title is misleading. Participants with obesity are excluded from analyses and the overall aim of the study is more concerned with longitudinal fluctations in weight.
Response 1: To clarify, subjects with a BMI of 18 kg/m2 to 40 kg/m2 at baseline were included in the generalized least square analyses (methods description line 111). Individuals who were obese, as defined as a BMI of 35 – 40 kg/m2, were included in the mixed models analyses examining changes to weight and BMI over time by level of avocado consumption (Table 3).
The marginal logistic regression analysis included only the subjects who were normal weight at baseline (line 161, Table 4).
The title has been changed to ‘Avocado intake, and longitudinal weight and body mass index changes in an adult cohort’.
Point 2: Introduction: Nice overview of implications of avocado consumption for weight management.
Response 2: Thank you
Point 3: Method: The Adventist Health Study – 2 is not well-explained. Seventh Day Adventists hold unique dietary beliefs that likely differentiates this group from the general population. For example, most Adventists do not eat pork and many are vegetarians. These differences are not acknowledged in the Method and only vaguely mentioned in the discussion.
Response 3: The methods (lines 61-66) have been amended to include a description of some of the peculiarities of the cohort along with a reference that provides a more detailed description: Butler, T.L.; Fraser, G.E.; Beeson, W.L.; Knutsen, S.F.; Herring, R.P.; Chan, J.; Sabate, J.; Montgomery, S.; Haddad, E.; Preston-Martin, S., et al. Cohort profile: The Adventist Health Study-2 (AHS-2). Int J Epidemiol 2008, 37, 260-265, doi:10.1093/ije/dym165.
Table 1 includes a report on the proportion of vegetarians versus nonvegetarians in the cohort. The descriptive results have been amended to state the proportion of subjects in the analytical sample that have been categorized as vegetarians (lines 174-175).
Point 4: Participants with obesity were excluded. Why?
Response 4: Subjects with obesity were not excluded from the analyses. Table 3 (previously table 2) has results from three categories of BMI (normal, overweight and obese). When we computed to odds of becoming overweight or obese over time, we included subjects who were normal weight at baseline (Table 4, previously table 2).
Point 5: Results: Appropriate analytic plan. Tables are somewhat cumbersome – perhaps authors could improve readability by not centering cells.
Response 5: We will follow-up with the editor (centered as per template).
Point 6: Discussion: Are there regional differences to consider? For example, if a preponderance of participants are from California (where avocados may be more available and rates of overweight/obesity are lower), this is an important confounder to consider.
Response 6: We acknowledge that subjects in California eat more avocado on average than outside of California (2.58 g/day versus 1.12 g/day). We did not find a substantial difference in BMI between those subjects in California than those residing outside the state (26.3 kg/m2 versus 26.7 kg/m2). We did sensitivity analyses on the mixed models analyses and logistic regression analyses with the addition of a region covariate variable which did not change the results. This has now been clarified in the results, lines 238-239, and 263-264.
Point 7: Other issues noted:
Line 66 – subject should be plural
Response 7: Amended (line 75).
Point 8-12: Line 68 – Is ‘dietary intake assessment’ a section heading?
Line 88 – section heading?
Line 101 – section heading?
Line 123 – section heading?
Line 149 – section heading?
Response 8-12: Yes. We will follow-up with the editor regarding these headings.
Point 13: Line 334 – y on ‘Health’
Response 13: Amended (line 367).
Round 2
Reviewer 1 Report
The authors responded well to most of my questions and improved their manuscript considerably. However, a few issues remain:
- Points 3-6, 8 and 13: Please add short statements to the manuscript to clarify these issues also for the reader.
- Point 15: I don't understand the rationale to calculate marginal means of BMI.
- Point 16: The differences in initial BMI indicate that the models in table 4 should also be adjusted for this variable, at least in a stepwise analysis.
- Point 17: So these results indicate that normal-weight non-consumers of avocado intake had a change of 0.79% in weight over five years, compared to 0.26% in normal-weight persons with high avocado intake? Is this really a clinically relevant difference, and not even in the range of measurement errors? The authors should mention these numbers in the main text, translate them into change in kg, and discuss this issue. Further, I would suggest to transform the beta coefficients in table 3 also to % weight change.
- Point 20: The authors should at least make their analysis code available, e.g. in an repository such as OSF (https://osf.io/), and add this link in the manuscript.
Author Response
Response to Reviewer 1 Comments (Second round)
Point 1: Points 3-6, 8 and 13: Please add short statements to the manuscript to clarify these issues also for the reader.
Response 1:
Point 3: lines 91-92, 102-104
Point 4: line 109
Point 5: lines 114-115
Point 6: lines 115-116
Point 8: lines 126-127
Point 13: line 62 and Table 1
Point 2: Point 15: I don't understand the rationale to calculate marginal means of BMI.
Response 2: Marginal means represent adjusted means. We hypothesized that the three avocado intake groups might have different participants’ profiles in demographics and energy intake. Therefore we adjusted the mean BMI values. Clarified on lines 129-130 and foot note of table 2.
Point 3: Point 16: The differences in initial BMI indicate that the models in table 4 should also be adjusted for this variable, at least in a stepwise analysis.
Response 3: The abstract, manuscript (lines 271-277, 289-298, 317-321, 379) and table 4 has been amended.
Point 4: Point 17: So these results indicate that normal-weight non-consumers of avocado intake had a change of 0.79% in weight over five years, compared to 0.26% in normal-weight persons with high avocado intake? Is this really a clinically relevant difference, and not even in the range of measurement errors? The authors should mention these numbers in the main text, translate them into change in kg, and discuss this issue. Further, I would suggest to transform the beta coefficients in table 3 also to % weight change.?
Response 4: We acknowledge that the changes in weight are small and of limited clinical relevance. For a 75 kg individual the difference in the percent weight change would translate to ~0.4 kg over 5 years between nonconsumers and high avocado consumers. Therefore, in terms of possible public health implications, the difference between nonconsumers and high consumers over a lifetime and at the population level, could be important. We have amended the manuscript – see lines 237-240, 318-321, 379-381.
We present estimated beta coefficients for the interaction between time and avocado intake because testing for the interaction indicates how the rates of weight/BMI change may be different between avocado consumers and non-consumers (lines 144-147). We acknowledge that the interpretation becomes somewhat awkward due to the log transformation of both sides of the regression equation. For this reason we also presented estimated weight/BMI changes in 5 years, using the same model, for avocado consumers and non-consumers (Lines 156-158; Table 3). We believe that comparing the estimated changes will be more informative to readers.
Point 5: The authors should at least make their analysis code available, e.g. in an repository such as OSF (https://osf.io/), and add this link in the manuscript.
Response 5: If the manuscript is accepted our plan is to submit the code to a repository along with a statement indicating where it can be found.